# Non-destructive in situ monitoring of structural changes of 3D tumor spheroids during the formation, migration, and fusion process

Ke Ning[1], Yuanyuan Xie[1], Wen Sun[2,3], Lingke Feng[1], Can Fang[4], Rong Pan[1], Yan Li[2,3]*, Ling Yu[1]*

[1]Key Laboratory of Luminescence Analysis and Molecular Sensing, Ministry of Education, Institute for Clean Energy and Advanced Materials, School of Materials and Energy, Southwest University, Chongqing, China; [2]Key Laboratory of Animal Biological Products & Genetic Engineering, Ministry of Agriculture and Rural, Sinopharm Animal Health Corporation Ltd, Wuhan, China; [3]State Key Laboratory of Novel Vaccines for Emerging Infectious Diseases, China National Biotec Group Company Limited, Beijing, China; [4]School of Computer and Information Science, Southwest University, Chongqing, China

**\*For correspondence:**
keliyanust@163.com (YL);
lingyu12@swu.edu.cn (LY)

## eLife Assessment

The ingenious design in this study achieved the observation of 3D cell spheroids from an additional lateral view and gained more comprehensive information than the traditional one angle of imaging. This extended the methods to investigate cell behaviors in the growth or migration of tumor organoids in a time-lapse manner and these extensions should be **important** to the field. The authors provide **compelling** evidence that the methods work as described.

**Abstract** For traditional laboratory microscopy observation, the multi-dimensional, real-time, in situ observation of three-dimensional (3D) tumor spheroids has always been the pain point in cell spheroid observation. In this study, we designed a side-view observation petri dish/device that reflects light, enabling in situ observation of the 3D morphology of cell spheroids using conventional inverted laboratory microscopes. We used a 3D-printed handle and frame to support a first-surface mirror, positioning the device within a cell culture petri dish to image cell spheroid samples. The imaging conditions, such as the distance between the mirror and the 3D spheroids, the light source, and the impact of the culture medium, were systematically studied to validate the in situ side-view observation. The results proved that placing the surface mirror adjacent to the spheroids enables non-destructive in situ real-time tracking of tumor spheroid formation, migration, and fusion dynamics. The correlation between spheroid thickness and dark core appearance under light microscopy and the therapeutic effects of chemotherapy doxorubicin and natural killer cells on spheroids' 3D structure was investigated.

## Introduction

Three-dimensional (3D) tumor spheroids have emerged as a powerful tool for studying tumor biology and drug response. These multicellular aggregates better recapitulate solid tumors' complex

microenvironment and heterogeneity than traditional 2D cell culture models (*Thoma et al., 2014*). 3D tumor spheroids exhibit gradients in oxygen, nutrients, and metabolites that mimic the pathophysiology of native tumors (*Derda et al., 2009*; *Huh et al., 2011*), making them valuable for investigating tumor growth, invasion, and response to therapies.

Visually, spheroids appear as translucent balls with a well-defined boundary and darker core when imaged by brightfield or phase contrast microscopy (*Amaral et al., 2017*; *Pan et al., 2023b*). Spheroid growth is characterized by an initial exponential phase followed by a plateau once the spheroid exceeds a critical diameter (typically around 500 μm) (*Amaral et al., 2017*; *Kim et al., 2021*; *Klowss et al., 2022*). Thickness measurements can detect this transition and provide a more sensitive readout of growth arrest than the length in the *x* and *y* directions (*Browning et al., 2021*). Characterizing the morphology and volume of 3D tumor spheroids is essential for understanding their growth kinetics and evaluating treatment effects. Directly measuring width/diameter and thickness enables more precise volume calculations that could compensate for spheroids' non-uniform shape and size (*Napolitano et al., 2007*).

However, light microscopy itself has limitations in quantifying the complex 3D structure of spheroids, especially the width and thickness of the spheroids simultaneously. The translucent nature of spheroids leads to blurred images that can cause overestimation of size and volume based on 2D measurements alone (*Costa et al., 2019*). Confocal imaging enables optical sectioning of spheroids to generate 3D reconstructions, and fluorescent labeling of cells allows direct measurement of thickness and diameter from cross-sectional views (*Raza et al., 2020*). However, light scattering and absorption limit imaging depth, making it challenging to resolve the core of larger spheroids (>300 μm) (*Fang et al., 2023*; *Pan et al., 2023a*; *Steinberg et al., 2020*). Optical coherence tomography is a non-invasive imaging technique that uses low-coherence light to create cross-sectional images of tissue structure, allowing measurement of thickness and diameter even in larger spheroids (*El-Sadek et al., 2021*; *El-Sadek et al., 2020*; *Abd El-Sadek et al., 2020*). Light sheet fluorescence microscopy (LSFM) is another emerging technique that illuminates samples with a thin sheet of light, enabling rapid 3D imaging of entire spheroids and measuring thickness and diameter from 3D reconstructions (*Eismann et al., 2020*; *Paiè et al., 2023*; *Shi et al., 2024*). However, LSFM requires specialized equipment and may be limited by light scattering in dense spheroid cores (*Andilla et al., 2017*; *Costa et al., 2019*).

Another challenge in the field is the lack of standardized protocols for measuring spheroid thickness and width across different imaging modalities and analysis software (*Froehlich et al., 2016*; *Koudan et al., 2020*; *Moraes et al., 2020*). Balancing spheroid thickness analysis while further observing changes in the sample's 3D structure is extremely challenging. Due to limited perspectives, current research tends to simplify the relatively regular morphology of spheroids into models that are centrally symmetrical or axis-symmetrical (*Senavirathna et al., 2013*). However, in addition to regular cell spheroid morphology, cell spheroids can undergo irregular changes. This inconsistency makes it difficult to compare results across studies and reproduce findings. Establishing guidelines for image acquisition, processing, and reporting is critical for advancing the field.

In this work, using conventional brightfield microscopy, we propose a side-view observation device to systematically study the correlation between bottom- and side-view of 3D tumor spheroids. We designate the bottom-view, visible through inverted microscope imaging, as the underside of the spheroid sample (*x-y* plane), where measurements such as width and diameter can be obtained. Images captured through the side-view observation device represent the side-view of the sample (*x-z* plane), from which measurements such as width, height/thickness, and contact angle with the *x-y* plane can be derived. First, the 3D spheroid side-view observation device was fabricated by assembling a cell petri dish, an agarose-micro-well array, and an optical module for observing. The device was applied to track the dynamics of 3D cell spheroid formation and cell migration from the spheroid. In addition, the two spheroids' fusion process and corresponding morphological changes at the bottom- and side-view were characterized to investigate the fusion dynamics. With the side-view observation, the correlation between the thickness of the spheroid and the dark core appearance under the light microscope examination was studied. Last, the killing effect of chemotherapy compound doxorubicin (DOX) and immune cell natural killer (NK) cells on the 3D structure of the spheroids were studied based on the side-view observation device.

# Results

## Validation of side-view observation achieved by placing a first-surface mirror adjacent to the 3D spheroids

The 3D spheroids were cultivated in agarose micro-wells within a standard petri dish. The assembled side-view observation device, featuring a 3D-printed handle and frame with an attached first-surface mirror, was placed directly on the microscope stage (*Figure 1A*). The magnets attract each other, allowing the divice to be positioned up and down along the petri dish lid. Magnet pairs embedded in the handle and frame allow flexible movement, enabling the mirror to be positioned near the sample by moving the handle along the petri dish lid (*Figure 1B*).

Bottom-view images were captured using an inverted microscope, providing measurements like diameter and width of spheroids (*x-y* plane, *Figure 1C*). Side-view images, facilitated by the mirror, offered additional measurements such as thickness, height, and contact angle (*x-z* plane, *Figure 1C*). The principle of non-destructive in situ observing the samples from both bottom- and side-view is illustrated in *Figure 1D*. For bottom-view, the standard microscope setup sufficed. For side-view, light from the source, either reflected by the sample or transmitted through it, was redirected into the microscope's objective by the first-surface mirror. This placement enabled capturing side profiles (supplementary information *Video 1*). *Figure 1E* shows the experiment setting of observing the bottom- and side-view of the samples. No modification to the microscope was done to achieve non-destructive in situ observation of the bottom- and side-view of cell spheroids at a low cost.

Maintaining image authenticity is a prerequisite for using a first-surface mirror to achieve side-view observation on an inverted microscope. Firstly, we tested image quality using a 0.3 mm round-type microscope calibration slide (ChenZheng Precision Tools, Suzhou, China). For bottom-view imaging, the sample was placed directly on the stage. For side-view imaging, it was cut and attached vertically to polyethylene foam double-sided tape (*Figure 1F*). Images captured with 4×, 10×, and 20× objective lenses showed no significant quality difference between bottom- and side-view observation. However, it was noted that the 40× objective lens' field has limitations in imaging (*Figure 1G, H*).

To study the impact of the distance between the sample and the mirror on image acquisition, a 0.03 mm line-type microscope calibration slide was fixed vertically to a base at a 90° angle to the horizontal plane (*Figure 1I*), and another microscope calibration slide to the bottom surface (*Figure 1J*). *Figure 1K* shows images captured between the mirror and the sample at 2, 3, and 4 mm distances. Side-view images remained similar across these distances. However, at 4 mm, the 40× objective could not focus due to exceeding its working distance (3.6–2.8 mm). The results demonstrated that as the sample was within the working distance of the objective lens, there was no significant change in the apparent size of the sample observing with 4×, 10×, 20×, or 40× objective lenses (*Figure 1L*).

For capturing spheroid images, the first-surface mirror was placed in a cell culture medium to maintain growth conditions. Typically, introducing different media into the light path causes refraction and potential chromatic aberration. For instance, light passing through the DMEM medium undergoes such refraction. To evaluate the impact of chromatic aberration on imaging resolution, we analyzed the offset values of the C-line (656.3 nm) and F-line (486.1 nm) using chromatic aberration analysis (*Figure 1M*). The analysis is as follows:

$$\Delta l = d \times NA \times \left[ \frac{1}{\sqrt{n_F^2 - NA^2}} - \frac{1}{\sqrt{n_C^2 - NA^2}} \right] \tag{1}$$

where *NA* is the numerical aperture, $\Delta l$ is the relative distance offset, *d* is the total length of the folded light path from the sample to the bottom of the petri dish, $n_F$ is the refractive index at 486.1 nm, and $n_C$ is the refractive index at 656.3 nm. Higher *NA* and longer *d* increase chromatic aberration, affecting observations (*Figure 1N*). However, as observed in *Figure 1O*, enabling the automatic white balance function during the imaging process can effectively minimize the chromatic aberration. *Figure 1P* shows that images at 4× and 10× in ddH$_2$O or DMEM had no significant differences compared to bottom-view images in air. Considering the distance between the sample and the mirror, and the culture medium-induced refraction, using the first-surface mirror for high magnification (40×) side-view imaging is not recommended.

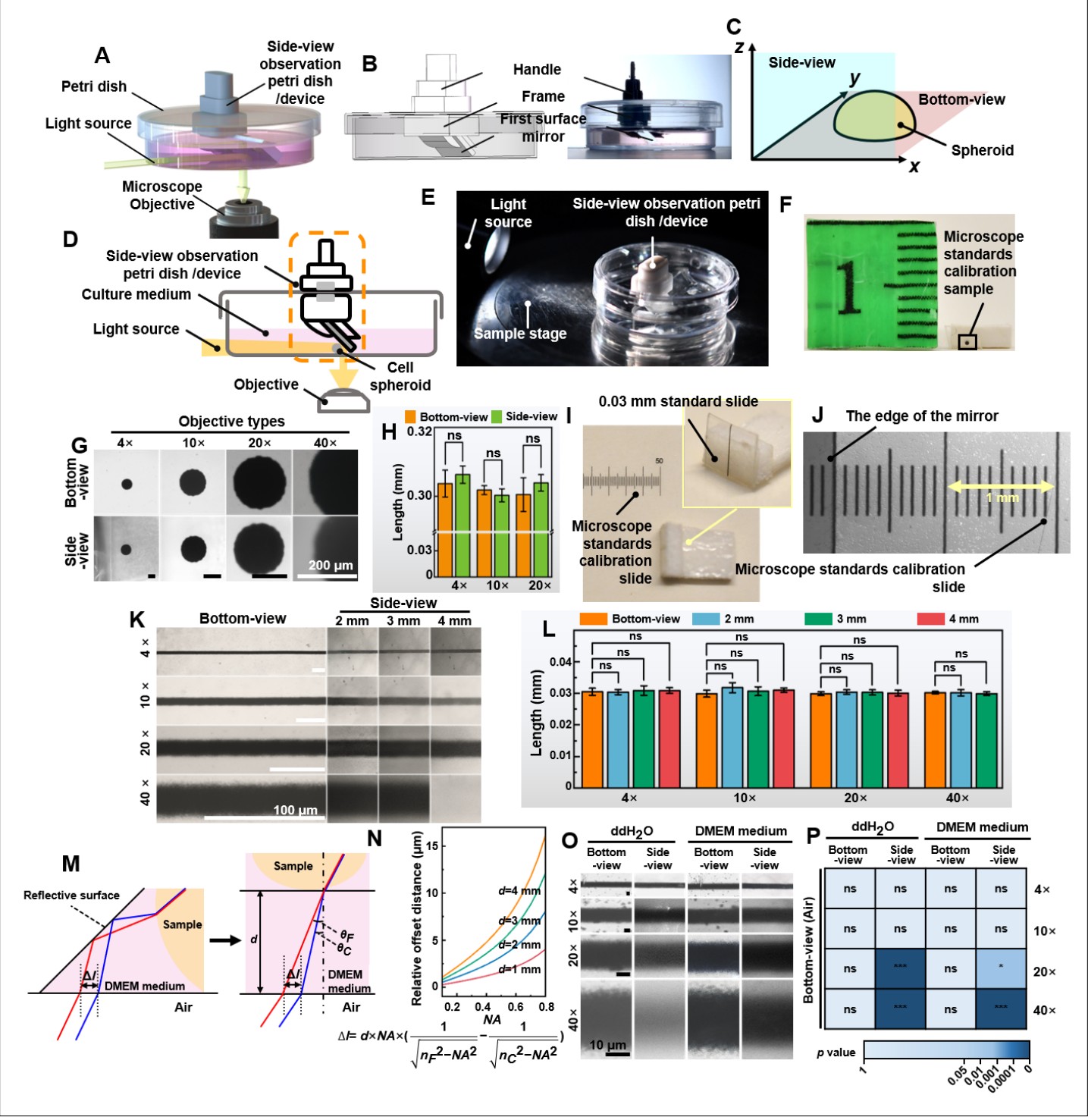

**Figure 1.** Validation of the non-destructive in situ observation of three-dimensional (3D) spheroids by the side-view observation device. (**A**) The illustration of the side-view observation device. (**B**) Exploded view diagram of the side-view observation device. (**C**) The definition of the bottom- and the side-view of a spheroid sample. (**D**) The working principle of the device. (**E**) The photo of the petri dish mounted on the sample stage of the microscope. (**F**) The 0.3 mm round-type microscope calibration slide for side-view observation. (**G**) The bottom- and side-view images of the 0.03 mm microscope calibration slide were captured with different objective lenses. (**H**) The length of the 0.3 mm dot measured from the bottom- and side-view captured images. (**I**) A 0.03 mm line-type microscope calibration slide was vertically fixed to a base positioned at a 90° angle to the start of another microscope calibration slide. (**J**) The photograph of the microscope calibration slide and the image of the edge of the first-surface mirror (gray part) by the microscope. (**K**) The images were captured with the distance between the mirror and the sample at 2, 3, and 4 mm. (**L**) The length of the 0.3 mm

*Figure 1 continued on next page*

*Figure 1 continued*

line-type calibration slide measured from the bottom- and side-view images captured with the distance between the mirror and the sample at 2, 3, and 4 mm. (**M**) The optical path in the cell culture medium and the chromatic aberration generated. (**N**) The fitting curve between *NA* and relative offset distance between 486.1 and 656.3 nm (Δ*l*) when *d* is 1, 2, 3, and 4 mm. (**O**) The bottom- and side-view images of 0.03 mm microscope calibration slide captured with 4×, 10×, 20×, and 40× objective lenses in ddH₂O and DMEM medium. (**P**) The length of the 0.3 mm line-type calibration slide was measured in ddH₂O and DMEM mediums through the bottom and side views (error bars = SD, *n* = 5).

## The side-view observation petri dish/device allows non-destructive, real-time observation of the bottom- and side-view of the 3D spheroid

Growing cells at agarose micro-wells or agarose-filling microplates is one of the effective methods to generate 3D spheroids (*Caprio and Burdick, 2023*; *Froehlich et al., 2016*). For imaging, our focus was on light emerging from the spheroid surface, not internal refraction or scattering. *Figure 2A* illustrates that during bottom-view imaging, the agarose micro-well directly contacts the bottom surface, requiring the image to pass only through the agarose layer to reach the microscope objective. For side-view imaging (*Figure 2B*), light passes sequentially through the agarose micro-well, culture medium (DMEM), reflects off a first-surface mirror, and finally enters the microscope objective. Therefore, the optical path through the agarose was the same for both bottom- and side-view imaging (with both bottom and wall thicknesses being 1.5 mm), resulting in identical refraction introduced by the agarose micro-well in both viewing modes.

Next, we studied the impact of agarose and culture medium on imaging quality. To support cell spheroid growth, we used micro-wells formed from 2 wt% agarose (*Pan et al., 2023a*). Absorption spectra of 1 wt% agarose, 2 wt% agarose, and DMEM culture medium (each 1 cm thick), were measured using a fiber optic spectrometer (PG2000-Pro-EX, Fuxiang Optics Co, Ltd, Shanghai, China), with ddH₂O as the control (*Figure 2C*). It was observed that both 1 and 2 wt% agaroses partially absorb light in the 400–600 nm range, while DMEM medium, containing phenol red as an indicator, had absorption peak at 420 and 550 nm (*Zhikhoreva et al., 2018*). The spectra of the aforementioned samples under the CEL-TCX250 (Xenon lamp light source) are shown in *Figure 2D*. The DMEM medium and agarose primarily absorbed light in the 400–600 nm range, with less absorption from 600 to 800 nm, causing the spheroids' images to appear magenta. During imaging, we used automatic white balance correction to minimize the impact of color on the cell images. Since this study focused mainly on the morphology of the spheroids rather than their color information, the introduction of the culture medium and agarose did not have a significant negative impact on the experimental results.

To optimize conditions for side-view imaging, we chose a xenon lamp for its broad wavelength coverage and adjustable angle. *Figure 2E* shows that a 90° angle between the lamp and first-surface mirror produced the best details of the side-view images, while angles of 45° and 0° resulted in uneven and poorly lit images. Maintaining the 90° angle, we examined the effect of varying distances between the light source and sample (*Figure 2F*). At 5 cm, internal structures (dark core) were clear, but uneven background illumination complicated edge distinction. At 10 cm, background uniformity improved, though overall illumination remained irregular, and the dark core was not resolved. At 15 cm, the sample background contrast was optimal, with clearly defined edges and discernible internal details, including the dark core. By adjusting the position of the sample stage, it became straightforward to locate and observe the side-view of spheroids within agarose micro-wells (supplementary information *Videos 2 and 3*). And the side-view observation petri dish/device can be used to photograph not only a single spheroid but also multi-spheroids samples (*Figure 2G*).

Collectively, for observing and imaging the side morphology of samples using the side-view observation petri dish/device, particularly for samples in culture medium, we recommend using a non-divergent light source with broad-spectrum

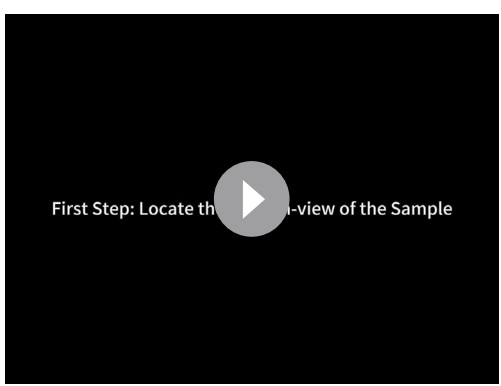

**Video 1.** Observation of the bottom- and side-view of a tumor spheroid using the side-view observation device.

https://elifesciences.org/articles/101886/figures#video1

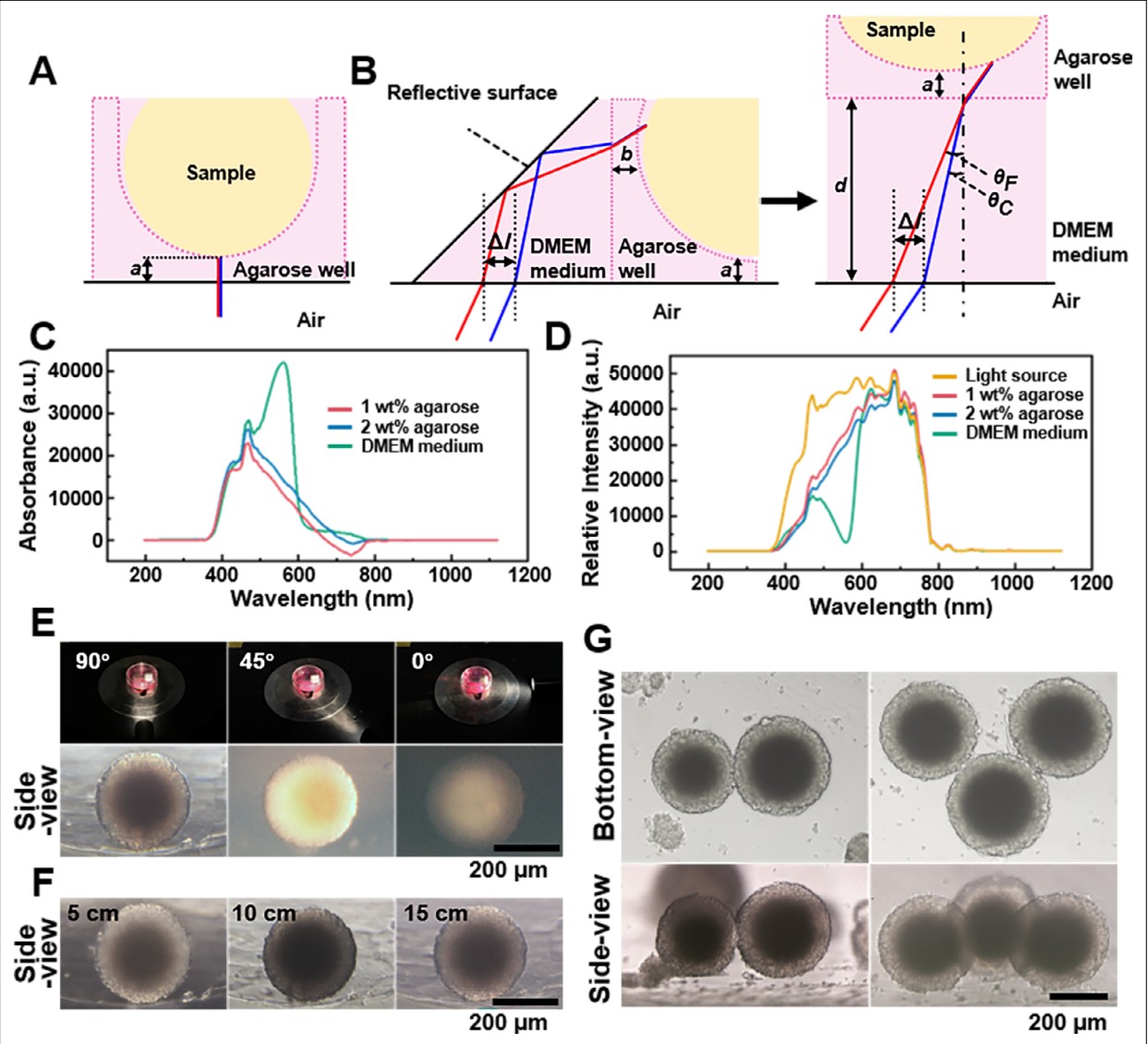

**Figure 2.** Spheroids' observation and imaging setup. (**A**) Light path of the bottom-view observation of the spheroid in the agarose micro-well. (**B**) Light path of side-view observing of the spheroid in the agarose micro-well. (**C**) The absorption spectrum of the 1 wt% agarose, 2 wt% agarose, and DMEM culture medium. (**D**) The spectra of the 1 wt% agarose, 2 wt% agarose, and DMEM culture medium using the CEL-TCX250 Xenon lamp light source. (**E**) Images quality display of spheroid under the 90°, 45°, and 0° lighting. (**F**) Images display of the spheroids under 5, 10, and 15 cm lighting. (**G**) Using the side-view observation petri dish/device can be used to photograph multi-spheroids samples.

coverage. This light source should be positioned approximately 15 cm directly behind the sample, forming a 90° angle with the mirror. Samples should be kept as close to the mirror as possible, with the distance between the sample and the mirror kept within the working distance of the objective lens. Considering the diameter of 3D spheroids ranges from 200 to 500 µm, side-view images were captured using 4× or 10× objective lenses in subsequent studies.

## Tracking the growth dynamics of spheroids from side-view observation

Tracking the growth images of cell spheroids enables further determination of their volume or size, which is significant for sample screening. However, conventional methods rely heavily on bottom-view images because it is challenging to monitor the side-view changes in situ. This study tracked the spheroid formation process in the agarose micro-well from a seeding density of $1 \times 10^4$ cells per well using the inverted microscope. Time-lapse images of the bottom- and side-view of the cells within the agarose micro-well, as shown in *Figure 3A*, record the morphological changes of

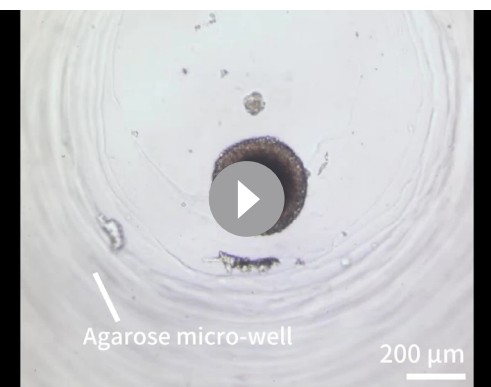

**Video 2.** Locating the side-view of the spheroid in the agarose micro-well through the bottom-view.
https://elifesciences.org/articles/101886/figures#video2

the cells during spheroid formation. Initially, the newly added human prostate cancer cells DU 145 scattered in the micro-well. After 3 hr of incubation, the cells clustered to form a ring structure. From the side-view, the thickness changes of the cell cluster were negligible, suggesting that the cells tend to aggregate at a similar planar. As the cells continued to aggregate, the cluster gradually became more compact and formed a round shape after 60 hr of incubation. However, the side profile of the same spheroid shows that the thin layer of the cell disk gradually packed, and as the *x–y* dimension decreased, the height/thickness of the cell aggregate increased (*Figure 3B*). We also placed the side-view observation petri dish/device on a live cell monitoring system (Moni-CyteTM B-100, Jiangsu Rayme Biotechnology, China) to capture a time-lapse video of the side-profile changes spheroid formation process of $1 \times 10^4$ DU 145 cells over 60 hr (Supplementary information *Video 4*). Through those time-lapse images/video, the width and height/thickness of the spheroid can be measured. As suggested in previous studies, when the height/thickness and width ratio of the cell aggregate approaches 1, it indicates that the spheroid structures have formed. In previous studies, spheroids had to be sealed or fixed in an agar block before the side-view morphology could be observed (*Pan et al., 2023a*). The side-view observation device allows for non-destructive tracking of the growth dynamics of the same spheroid in both bottom- and side-view, which enables the depict the 3D structural characters of cell aggregates.

## Characterization spheroids 3D structure changes during spheroid migration

Cell migration from or leaving tumor spheroids is crucial for the dissemination of solid tumors and the occurrence of secondary metastases, both of which are life-threatening. Previously, spheroids were placed on a substrate, and their migration capability was characterized by cell migration distance (*Carvalho et al., 2022*; *Kim et al., 2022*). This study placed tumor spheroids on a petri dish for a cell migration assay. As the bottom-view morphology shown in *Figure 4A*, cells extruded from the spheroid, forming sprouts. With prolonged culture time, cells from the spheroid periphery migrated outwards, forming a single-layer cell region surrounding the spheroid. The migration distance is the difference between the total width (spheroid + migrating length) and the inner width (spheroid only). As shown in *Figure 4B*, the distance occupied by the migrating cells expanded from 0 to 2894.59 ± 127.08 μm after 100 hr of migration. It was important to note that the average inner width only decreased by 23.87 μm (a change rate of 6.8%).

Simultaneously, with the assistance of the side-view observation device, variations in the spheroid's height and contact angle during the migration process were recorded (*Figure 4C* and supplementary information *Video 5*). *Figure 4D* elucidates the measurement benchmarks for the side-view images. The contact angle between the petri dish surface and the spheroid was measured using the angle function of ImageJ. As shown in *Figure 4E*, when the spheroid was placed on the petri dish, the thickness of the spheroid was 242.34 ± 2.85 μm, and the contact angle between the surface and the spheroid was 0°. After 28 hr of migration, the thickness of the spheroid gradually decreased to 161.11 ± 13.08 μm, and the contact

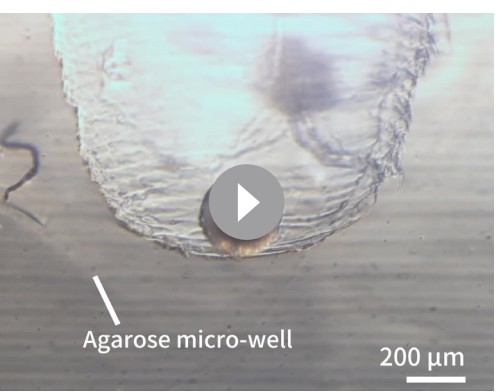

**Video 3.** Recording the side-view images of different spheroids in the agarose micro-well.
https://elifesciences.org/articles/101886/figures#video3

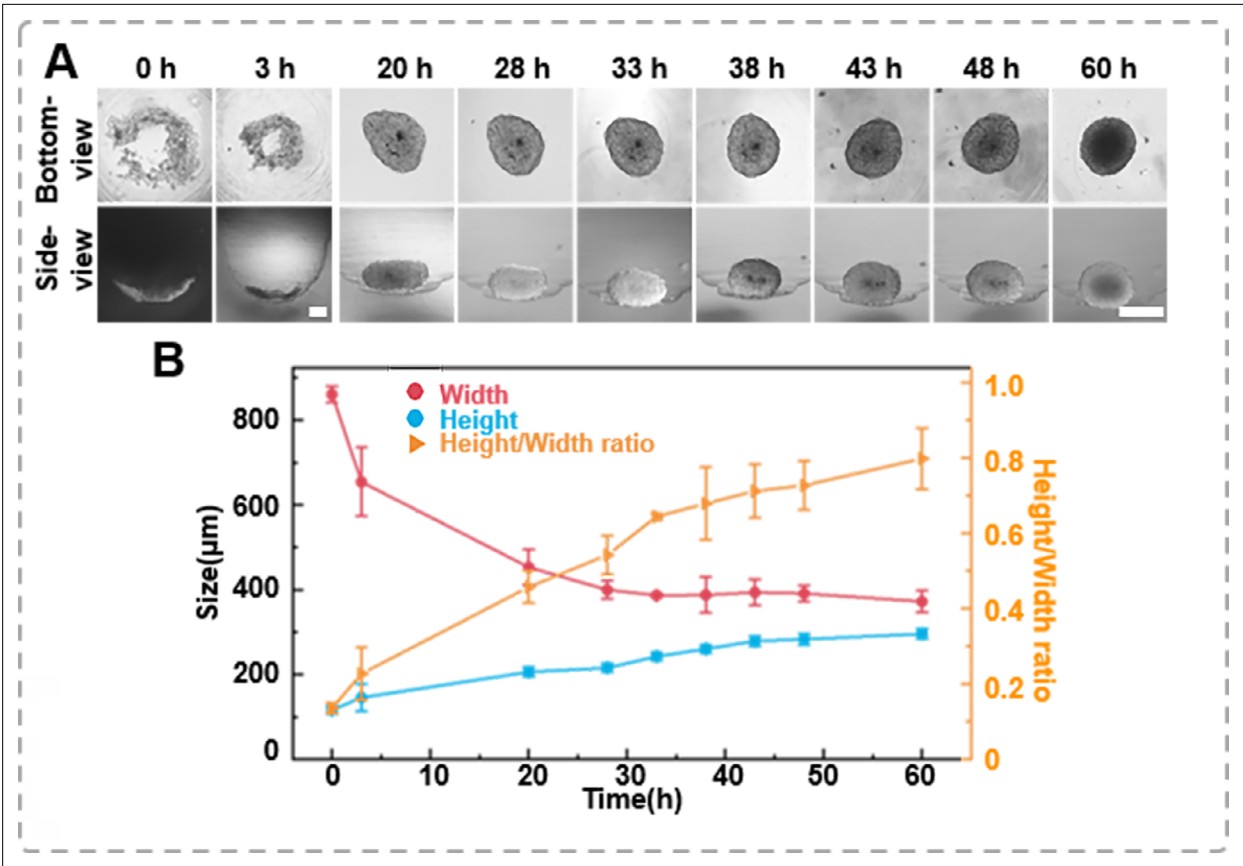

**Figure 3.** The formation dynamics of the spheroids. (**A**) Time-lapse images of DU 145 spheroid formation in an agarose well (1 × 10⁴ cells/well, scale bar = 100 μm). (**B**) Changes in height and width during the growth process of the spheroids, as well as variations in height-to-width ratio.

angle changed to 98.52 ± 7.57°. Throughout 100 hr of migration, the thickness and the contact angle of the spheroids' changed to 131.33 ± 8.34 μm and 172.08 ± 3.75°, respectively. Furthermore, from the side-view images, the ImageJ's measurement function revealed that the dark core area was 46,125.40 μm² at 0 hr and only decreased to 43,302.44 μm² at 100 hr. Based on the side-view images, we calculated the difference between the height of the outer spheroid and the height of the internal dark core, denoted as ΔH. It was found that the height of the dark core decreased by 57.63 μm, while the total spheroid height decreased by 111.01 μm after 100 hr of migration. The dark core height ratio increased from 0.61 to 0.69, suggesting that the proliferating cells at the outer region of the spheroid were the prominent participants in the migration process (*Figure 4F*). Combining the bottom- and side-view images, it was found that as the average width increases, the *z*-direction height of the spheroids decreases, leading to a gradual decrease in the height-to-width ratio of the cell spheroids and an increase in the contact angle of the cell spheroids with the petri dish.

## Association between the thickness of the spheroid and the appearance of the dark core of the 3D spheroid

A dark core within 3D cell spheroids is a phenomenon commonly observed. Biologically, the dark core is believed to represent necrotic or quiescent zones arising from limited diffusion of oxygen and

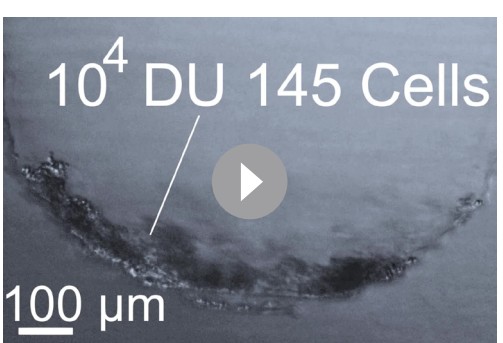

**Video 4.** Non-destructive in situ observation of the thickness changes during DU 145 cell spheroids formation process.

https://elifesciences.org/articles/101886/figures#video4

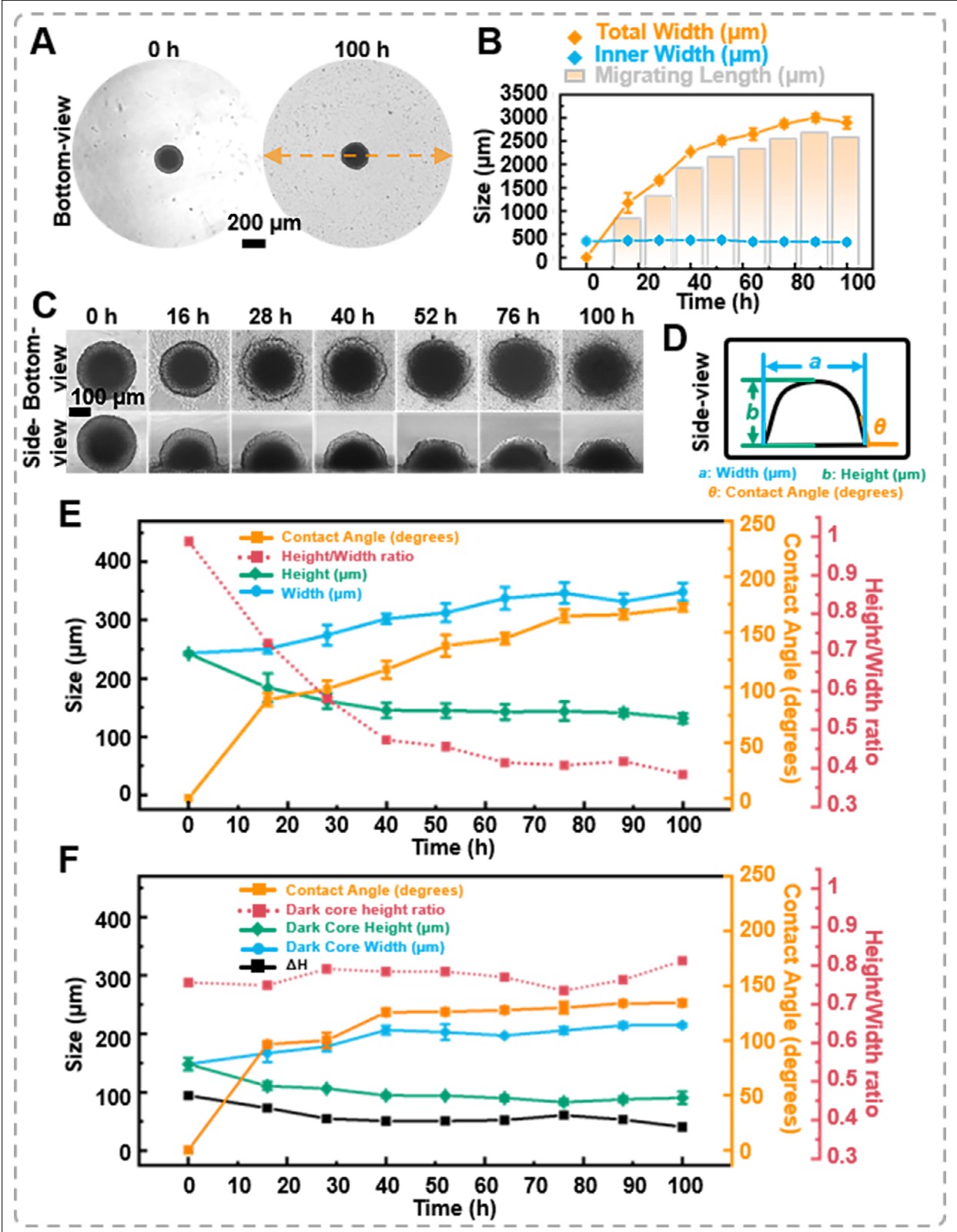

**Figure 4.** The migration of the spheroids. (**A**) The bottom-view images of the spheroid in the petri dish, cells extrude from the spheroid, forming sprouts. (**B**) Statistics on the changes in total width, inner width, and migration distance of spheroids during migration process. (**C**) Time-lapse bottom- and side-view images of DU 145 spheroids during migration process. (**D**) Definition of the measured data points based on the side-view images of spheroids. (**E**) Trend of morphological changes in the outer layer of spheroids during the migration process. (**F**) Trend of morphological changes in the dark core of spheroids during the migration process (error bars = SD, *n* = 3).

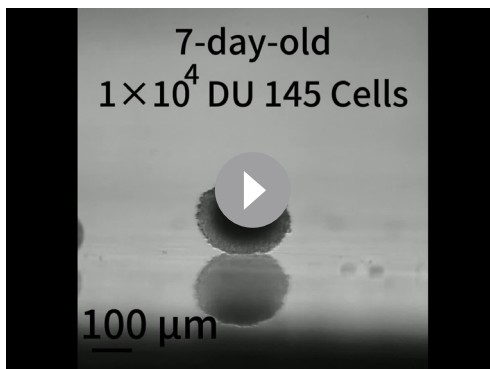

**Video 5.** Non-destructive in situ observation of the thickness changes during DU 145 cell spheroids migration process.
https://elifesciences.org/articles/101886/figures#video5

nutrients within the spheroid. In addition, it is important to notice that, optically, the depth of light penetration in microscopy can be hindered by the spheroid's thickness, leading to decreased signal intensity and diminished image quality from deeper regions. This effect is further amplified in larger spheroids by the increased number of cellular layers and a dense extracellular matrix, which scatter light and obscure core details. The side-view function of the proposed device allows tracing the formation of the visually distinguishable dark core to investigate the relationship between the dark core, spheroid size, and the optical limitations of light microscopy using DU 145 cell spheroids with different initial seeding densities ($2.5 \times 10^3$, $5 \times 10^3$, $1 \times 10^4$, and $2 \times 10^4$ cells/well). The appearance of the dark core and its correlation with spheroid thickness were analyzed using the bottom- and side-view of the spheroids.

As *Figure 5A* shows, it was found that the time to dark core formation was inversely related to the initial seeding density. For example, spheroids initiated with $2.5 \times 10^3$ cells exhibited a dark core between 48 and 60 hr, achieving an average z-direction growth rate of 4.18 µm/hr. These spheroids reached a nearly spherical shape with an average height of 250.67 µm and a maximal height-to-width ratio of 0.99 by 60 hr. With an increase in seeding density to $5.0 \times 10^3$ cells, the dark core appeared earlier (38–43 hr), and the spheroids achieved a height-to-width ratio of 0.82 by 43 hr and approached a spherical shape by 60 hr, with a width between 270 and 290 µm. At a seeding density of $1.0 \times 10^4$ cells, the dark core developed between 28 and 33 hr, with a z-direction growth rate of 7.35 µm/

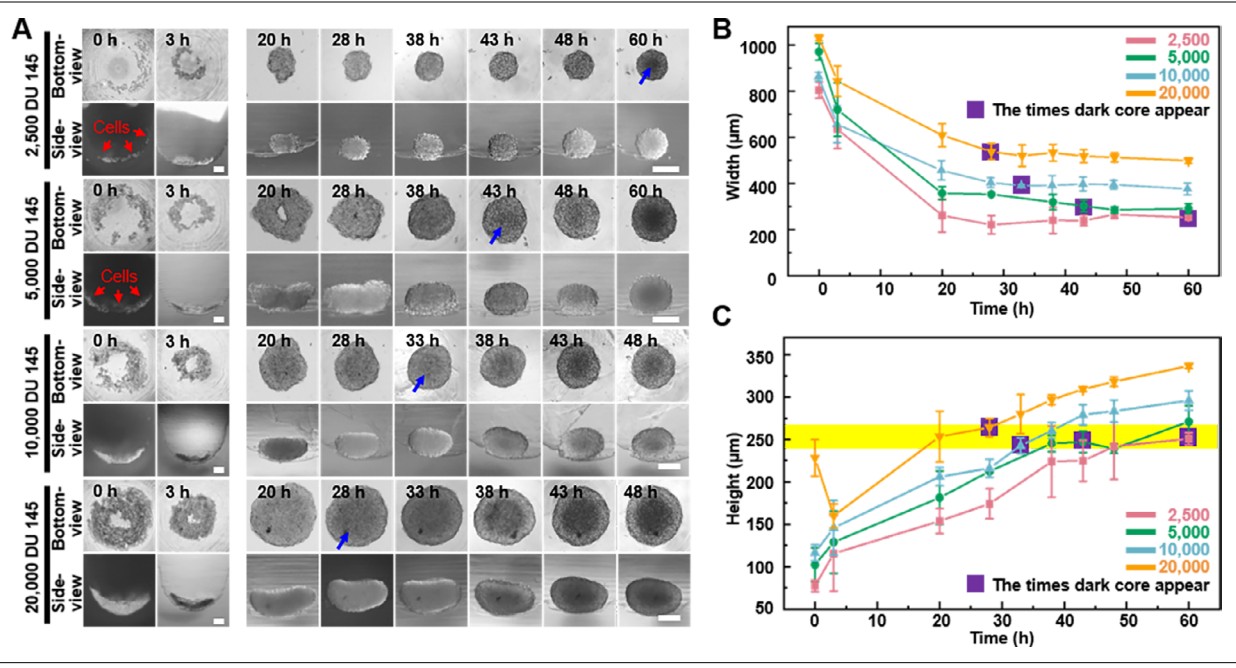

**Figure 5.** The dynamics of dark zone formation within the three-dimensional (3D) spheroid. (**A**) Time-lapse bottom- and side-view images of DU 145 spheroids formed from different initial cell concentrations ($2.5 \times 10^3$, $5 \times 10^3$, $1 \times 10^4$, and $2 \times 10^4$ cells/well). Blue arrows indicate the appearance of observable dark cores. (**B**) Relationship between spheroid width changes and dark core appears at different initial cell concentrations analyzed through bottom-view images. (**C**) Relationship between spheroid height changes and dark core appears at different initial cell concentrations analyzed through side-view images (the yellow stripe indicates the height of 250 ± 15 µm) (error bars = SD, *n* = 3).

hr. The spheroids presented a height-to-width ratio of 0.54 at 33 hr, evolving into a flattened spherical shape by 60 hr because the side-view images indicate an elongated, bean-shaped profile. The highest seeding density of $2.0 \times 10^4$ cells resulted in the earliest dark core formation (20–28 hr), with a $z$-direction growth rate of 9.43 µm/hr. These spheroids displayed the most pronounced elongation, with a height-to-width ratio of 0.49 at 28 hr, further flattening by 60 hr. We analyzed the relationship between dark cores' appearance and width using conventional bottom-view images of spheroids (*Figure 5B*). It is evident that with fewer seeding densities, the width of the spheroids gradually decreases over time, while the appearance of the dark core occurs later. The purple square highlight indicates the time points when the dark core appeared. Based on the trend observed in the images, we can predict that at lower seeding densities, the time for the appearance of dark cores in the spheroids will be longer, and vice versa. However, irrespective of the seeding densities, the $z$-direction height of the spheroids at the onset of dark core formation remained relatively consistent, approximately 250 ± 15 µm (the yellow-colored zone in *Figure 5C*), suggesting that there is a correlation between the formation of dark cores in spheroids and the $z$-direction height of the spheroids.

## Characterization spheroids 3D structure changes during spheroid fusion

Based on observing the spheroids' formation and migration, we hypothesize whether the side-view observation device facilitates the observation of more intricate cellular spheroids' structural changes through side profile. Fusion is a crucial step for tissue development. In tissue engineering, 3D spheroids fusion is vital in fabricating micro-tissues (*Lindberg et al., 2021*; *Laschke and Menger, 2017*). However, the fusion process of spheroids exhibits a highly random situation, making it challenging to maintain the fusion state while observing it in situ to reconstruct the structural changes during spheroid fusion. Herein, the side-view device was applied to conduct in situ observations of the fusion process of cell spheroids. Spheroids of $1 \times 10^4$ DU 145 cells per well were cultured for 7 days in agarose micro-wells for fusion assay. As the bottom-view images shown in *Figure 6A–i*, the two spheroids contact each other and gradually merge at the contact region. The contact angle, contact length, and doublet length value can all be retrieved from time-lapse images (*Figure 6B*). First, the contact angle between spheroids increased from 30° to 45°. Next, the two spheroids' contact length (neck) increased with co-culture time, with the most significant change occurring within the first 3 hr. The doublet length of the fusion formed by two spheroids decreased by 22.5% from 453.87 to 367.19 µm after 48 hr of fusion. Though the bottom-view depicts the $x$–$y$ dimension changes of the spheroids during the fusion process, predicting the phenome occurring on the side profile of the spheroid is challenging, especially for those cell aggregates without ideal spherical structures (the red arrow denoted in *Figure 6A–ii*). With the assistance of the side-view device, it was found that the two spheroids contacted and formed symmetric necks that gradually fused (*Figure 6A–i*). As shown in *Figure 6B*, the contact length between two spheroids progressively approaches the $z$-direction thickness, and the contact angle between the two spheroids reached 171.10 ± 13.94° after 100 hr of fusion. The green box in *Figure 6B* highlights that the fusion process of two spheroids can be transformed into a process where the contact length approaches the height. Thus, supplemented by the side-view perspective, modeling, and reconstruction of the cell spheroid fusion process become feasible (*Figure 6C*). The vertical lines in the figure indicate the position where the arc representing the contact angle at the fusion site of the cell spheroids tangentially intersects both spheroids. A wider width between the two vertical lines indicates a higher fusion process between the two spheroids. Additionally, we observed an intriguing phenomenon: if the initial cell spheroid morphology is not entirely regular or spherical (the red arrow denoted in *Figure 6A–ii*), the side-view profiles enable to track the changes in the particular region except the fusion region. Therefore, the reconstruction from the side-view perspective can be more comprehensive.

## Evaluate the impact of antitumor therapeutic reagents on the spheroid integrity

Spheroids are important in vitro cell models for drug testing, and the impact of drug effects is typically evaluated by the morphology and size of the spheroids (*Lu et al., 2015*). In this study, we investigated the effects of DOX, one of the major chemotherapy reagents currently clinically used, and NK-92 cells, a type of cytotoxic lymphocyte, on the 3D structure of DU 145 prostate cancer spheroids.

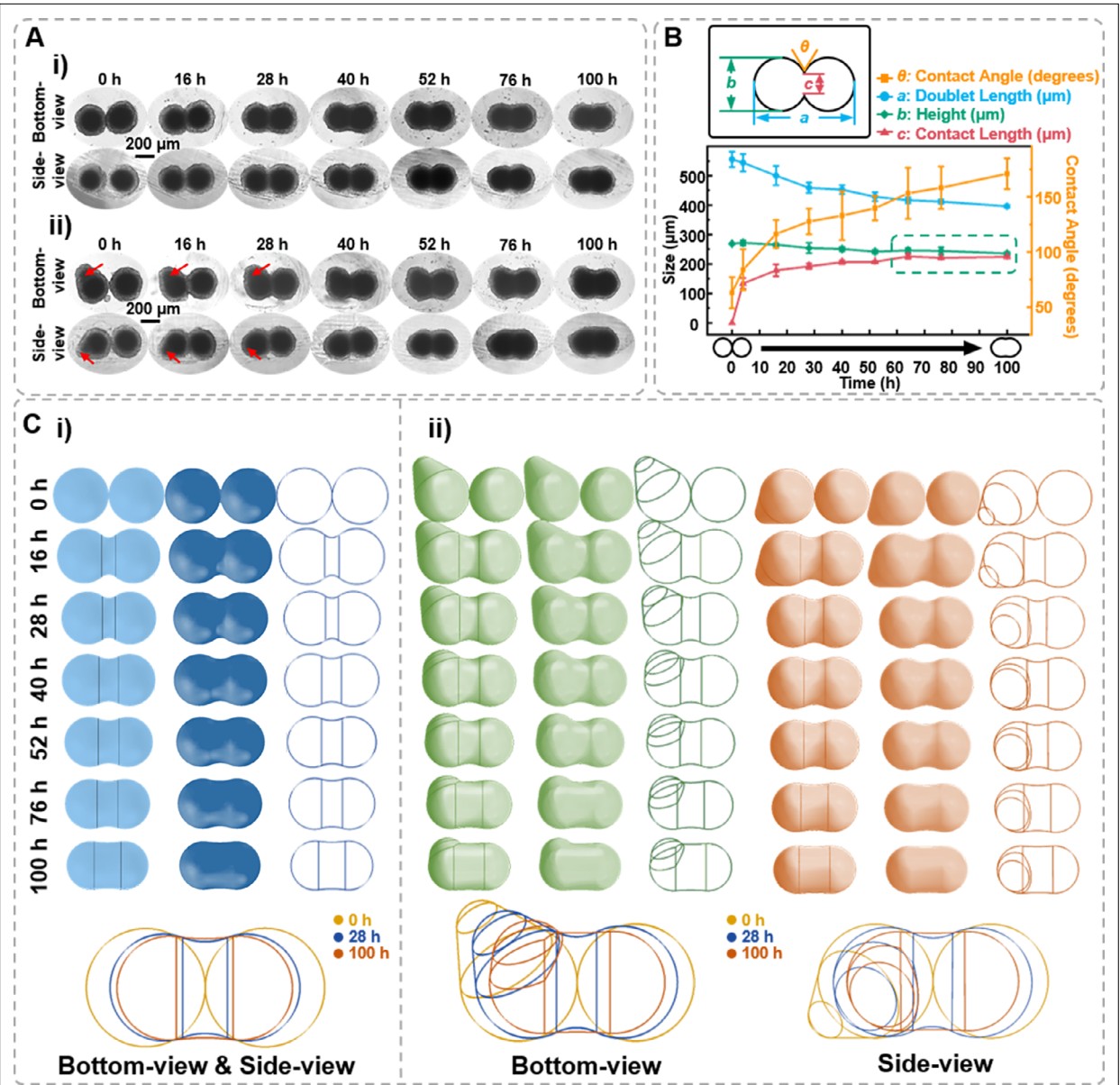

**Figure 6.** Characterization of the spheroids fusion process. (**A**) Time-lapse bottom- and side-view images of fusion body formed by two 7-day-old DU 145 cell spheroids with ideal ball-like structure (**i**) and irregular cell spheroid (**ii**). Red arrows point to the protruding part of the cell spheroid. (**B**) Definition of the measured data points based on the side-view images of fusion spheroids and the changes in doublet length, height, contact angle, and contact length during the fusion process. (**C**) Three-dimensional spheroid fusion process modeling using bottom- and side-view images of ideal ball-like structure (**i**) and irregular cell spheroid (**ii**). The vertical lines on the model surface illustrate the position where the arc tangentially intersects both spheroids (error bars = SD, $n$ = 3).

First, DU 145 spheroid (1 × 10⁴ cells/spheroid) was cultured for 3 and 7 days and treated with 50 µg/mL DOX for 96 hr. The spheroids aged for 3 days exhibited a relatively smaller dark core, with a diameter of 100.67 ± 24.65 µm, and the dark core of 7 days spheroids is 168.32 ± 5.98 µm. Time-lapse imaging (*Figure 7A*) revealed that with extended DOX treatment, the outer region of the 3-day-old spheroids gradually lost its smoothness, as observed in both bottom- and side-view. Furthermore, the visually dark core area was significantly enlarged. Similar morphological changes were noted in the 7-day-old spheroids. Meanwhile, it was found that DOX affects the height and width of the spheroids, causing them to expand within a certain period and then decrease in size as time progresses (*Figure 7B*). It is worth noting that despite the significantly changed surface smoothness, the ratio

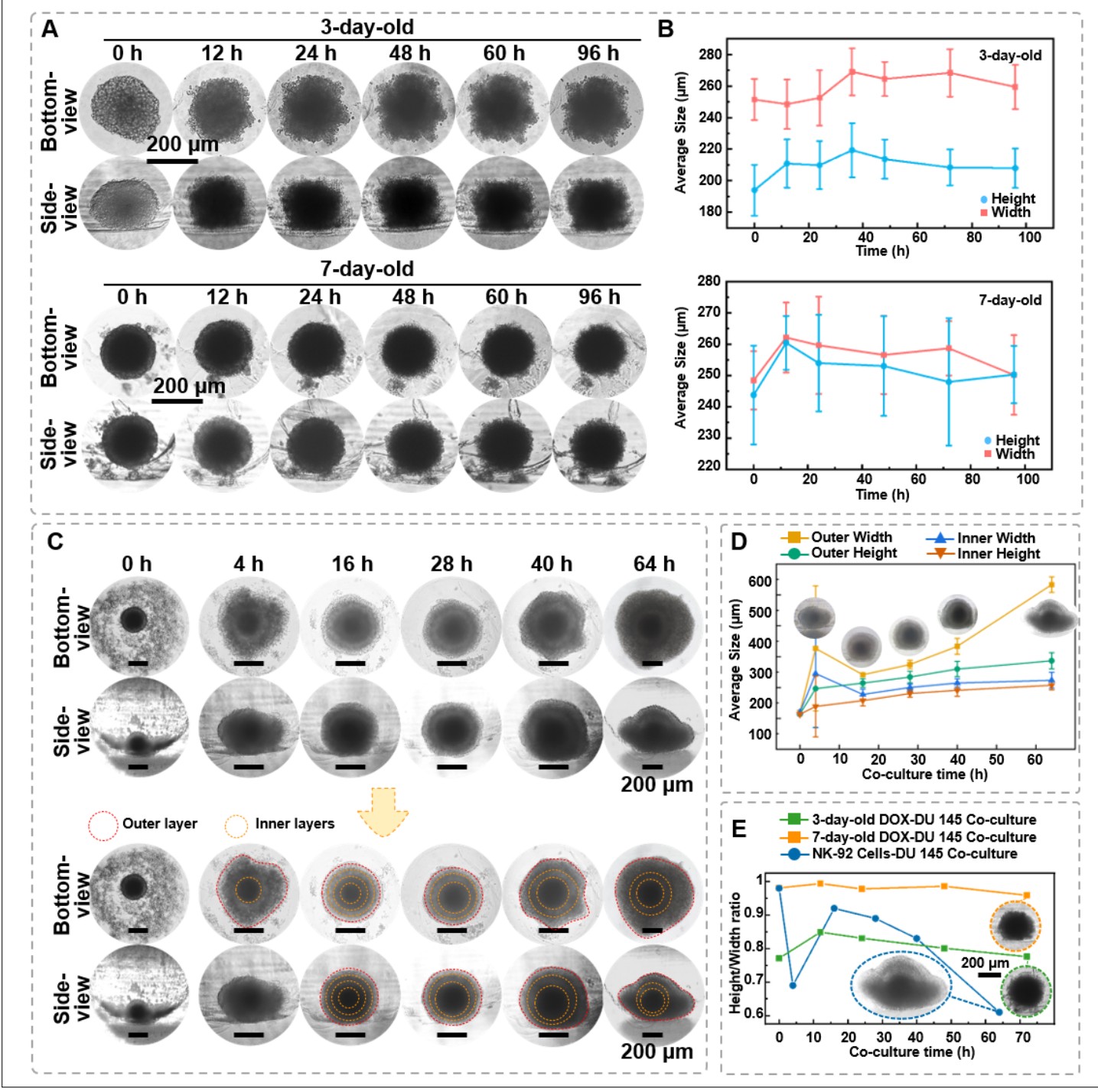

**Figure 7.** The in situ observation of DU 145 spheroids under the killing effect of doxorubicin (DOX) and natural killer (NK) cells. (**A**) Time-lapse bottom- and side-view images of 3- and 7-day-age DU 145 cell spheroids ($1 \times 10^4$ cells/spheroid) under DOX (50 µg/mL) treatment. (**B**) The changes of DU 145 spheroids' width and height upon DOX treatment. (**C**) Time-lapse bottom- and side-view images of 7-day-old DU 145 cell spheroids ($1 \times 10^4$ cells/spheroid) co-cultured with $1 \times 10^4$ NK-92 cells. (**D**) Changes in internal and external width and height of 7-day-old DU 145 cell spheroids ($1 \times 10^4$ cells/spheroid) co-cultured with NK-92 cells. (**E**) The different performances of DOX and NK-92 cells killing on the overall height-to-width ratio of the spheroids ($n$ = 3, error bars = SD).

of overall height-to-width remained largely unchanged, suggesting the spheroid's overall spherical structure was preserved.

Next, we also explored the role of NK-92 cells, a type of cytotoxic lymphocyte critical in the immune system's defense against tumors, as model immune cells. As depicted in *Figure 7C*, the 1 × 10⁴ NK-92 cells spheroid in the micro-well at 0 hr of the co-culture. After 4 hr of co-culture, NK-92 cells gathered around and enveloped the DU 145 spheroid as increased diamension in both bottom and side profiles. After 16 hr of co-culture, unlike the distinct boundary between dark core and translucent zone, referring to necrotic and proliferating regions of the spheroid, observed before co-culture, four optically different regions could be distinguished from both bottom- and side-view, suggesting that NK-92 cells infiltrated the DU 145 spheroids from all directions. At this time point, the overall height and width of the spheroid were 314.47 ± 13.85 and 341.20 ± 5.81 μm, respectively. Thus, the height-to-width ratio was close to 1, suggesting a spherical structure. After 28 hr of co-culture, only three optically different regions could be distinguished, and the distinction between the inner two regions became less obvious than at 16 hr. Moreover, the side-view images found that the spheroid appeared to collapse as the height-to-width ratio decreased to 0.89. However, measurements indicated expansion in the *x*–*y* and *x*–*z* directions, with overall growth rates of 9.78% and 6.22%, respectively. After 40 hr of co-culture, the collapse of the outer layer was apparent, and the height-to-width ratio was 0.83. At the same time, the spheroid's overall size continued to increase, with growth rates of 15.74% and 7.73% in the *x*–*y* and *x*–*z* directions, respectively. After 64 hr, the outer cell layer had completely collapsed. Cells had detached onto the bottom of the agarose well, losing the spherical shape (*Figure 7D*). By correlating the size growth rates observed in side-views with corresponding time points, the significant increase in the width at 4 hr was attributed to the external aggregation of NK-92 cells on the spheroid. Over time, NK-92 cells gradually envelop the spheroid without a noticeable change in the *z*-direction. From 20 to 60 hr, both bottom- and side-directions exhibited growth until the upper region collapsed downwards at 64 hr.

Analyzing the 3D structural changes of spheroids under DOX and NK-92 cells treatment indicates that through different mechanisms, antitumor agents result in distinct morphological features of the spheroid. *Figure 7E* compares the height-to-width ratio of the spheroids treated with DOX and NK-92 cells for 64 hr. The cytotoxic effect of NK cells on spheroids resulted in a 37% decrease in the height-to-width ratio, while there was only a 0.64% change for 3-day-old spheroids treated with DOX and a 2.24% change for 7-day-old spheroids treated with DOX. Therefore, viewing the spheroids from different angles, such as bottom- and side-view, would provide precise structural changes in the drug-induced killing effects.

## Discussion

We developed a side-view observation device using 3D-printed components and a first-surface mirror, which costs less than $1 to produce, to address the current limitations in observing 3D cell spheroids. First, the proposed side-view observation device was validated for observing the spheroid formation and migration process from both bottom- and side-view. As shown in *Figure 3*, side-view observation provides detailed insights into the spheroid formation, including the 3D structure of the spheroid and its height at a given time point. Past studies have often assumed spheroids to be centrally symmetrical (*Piccinini et al., 2015*), elliptical (*Murphy et al., 2011*), or spherical (*Senavirathna et al., 2013*). It is evident that as the spheroid grows, its morphology does not directly conform to a regular shape (*Figure 3A, B*). In situ tracking and imaging from a side-view perspective can facilitate modeling to reconstruct the real shape of the spheroid accurately and enable researchers to gain a more comprehensive understanding of the 3D structure and growth kinetics of spheroids without disrupting their natural development.

This device offers a unique opportunity to study the 3D aspects of cell migration from tumor spheroids (*Figure 4C*). The side-view images allow us to analyze the interaction between the spheroids and substrate by measuring the contact angle. Moreover, the changes in the dimensions and area of the dark core were less intense than the overall changes of the spheroid, suggesting that the proliferating cells at the outer region of the spheroid were the main participants in the migration process. To the best of our knowledge, this is the first time the side-profile changes of the spheroid during a migration assay have been depicted. By monitoring both bottom and side changes, researchers can gain valuable insights into the spatial distribution of migrating cells and the dynamics of the migration process

to elucidate the mechanisms underlying tumor cell invasion and metastasis, potentially leading to the development of new therapeutic strategies targeting these processes.

The observation of the dark core in the spheroid highlights the presence of a necrotic or quiescent region, which is a common feature of larger tumor spheroids (*Browning et al., 2021*; *Lindberg et al., 2021*; *Laschke and Menger, 2017*). As demonstrated in *Figure 5*, we observed that spheroids formed at different cell seeding densities, regardless of whether they approached a height-to-width ratio close to 1, began to develop a dark core at an overall height of approximately 250 ± 15 µm. This consistency suggests that the timeframe for dark core formation is primarily determined by the increasing *z*-direction height, with a critical threshold of approximately 250 µm may indicate the limit of effective oxygen diffusion, beyond which cellular functionality becomes compromised. The observation aligns with previous findings that suggest a maximum viable spheroid diameter of 500 µm, beyond which central necrosis becomes inevitable (*Pan et al., 2023a*). This clear observation was not achievable using only bottom-view images, emphasizing the importance of real-time monitoring and 3D structural analysis of spheroids through side-view observation. For the spheroids' fusion process, conventional observation using only bottom-view imaging might miss the critical structural changes on the other side of the fusion process (*Figure 6A*). However, with the side-view observation device, we could assist in the observation of 3D structural changes from the side, thus in situ monitoring of irregularly shaped spheroids, tissues, and organoids from multiple angles become possible.

Finally, we compared the cytotoxic effects of two different approaches, DOX and NK-92 cells, on spheroids using the side-view observation device. *Figure 7* shows the distinct outcomes of DOX treatment versus NK cell-mediated cytotoxicity: DOX treatment does not significantly alter the 3D morphology of the spheroids, whereas NK cell-mediated cytotoxicity notably affects the overall height-to-width ratio of the spheroids. DOX exerts its effects by disrupting DNA and RNA synthesis, inducing ROS-related apoptosis, and triggering cellular senescence, among other mechanisms (*Agudelo et al., 2016*; *Kong et al., 2022*). These actions lead to morphological changes such as nuclear condensation, fragmentation, and cell shrinkage. The observed morphological features from both the bottom- and side-view suggest that DOX may induce a form of cell death in tumor spheroids without disrupting their overall 3D structure (*Figure 7A*). On the other hand, as shown in *Figure 7C, D*, NK-92 cells gathered around the tumor spheroids and gradually infiltrated into the spheroids, leading to the expansion of the overall size of spheroids. Moreover, the co-culture of NK cells and tumor spheroids resulted in significant structural changes, as the height-to-width ratio decreased from 0.98 to 0.61 after 100 hr of co-culture. Considering that NK-92 cells exert cytotoxicity by forming pores in the target cell membrane, leading to membrane disruption and cell lysis, the side-view profile indeed captured the gradually demolished spheroid's 3D structure. Relying solely on bottom-view observations of co-cultured spheroids may not adequately predict side-profile changes or the overall 3D structure, potentially resulting in volume prediction inaccuracies. Therefore, by accurately observing the responses of spheroids to different cytotoxic agents, researchers can design more precise drug induction protocols and develop more effective strategies for tumor eradication.

The proposed device is compatible with inverted microscopes, allowing for in situ bottom- and side-view observation of 3D cell spheroids, a capability not previously reported. Compared to existing side-view microfluidic devices (*Lee et al., 2020*), our device offers lower production costs and is easier to manufacture. Additionally, the device features magnetic connections for easy positioning adjustment to observe multiple samples (*Video 3*) and convenient disassembly for reuse and sterilization purposes. To further improve the capability of the side-view device to analyze the sophisticated detail of 3D spheroids, such as the visually optical distinct layers in NK-tumor spheroids co-culture (*Figure 7D*), we are considering two avenues for optimization: improving the image capture process and enhancing image quality. One approach involves adding lenses in front of the first-surface mirror. At the same time, the other entails designing complementary light sources to reduce light scattering within the spheroids during imaging. Once the image capture process is further enhanced, we can leverage machine learning to analyze the side-view morphology of the spheroids and organoids to predict their future growth patterns, potentially saving considerable screening time in preclinical drug testing.

## Materials and methods
### Reagents

**Key resources table**

| Reagent type (species) or resource | Designation | Source or reference | Identifiers | Additional information |
|---|---|---|---|---|
| Cell line (human prostate cancer cell) | DU 145 | Pricella Biotechnology | CL-0075 (RRID:CVCL_0105) | |
| Cell line (human natural killer cell) | NK-92 | Pricella Biotechnology | CL-0530 (RRID:CVCL_2142) | |
| Chemical compound, drug | DMEM culture medium | Gibco | C11995500BT | |
| Chemical compound, drug | Fetal bovine serum | Bio-channel | BC-SE-FBS01 | |
| Chemical compound, drug | Penicillin–streptomycin solution | Beyotime Biotechnology | C0222 | |
| Chemical compound, drug | NK-92 special medium | Pricella Biotechnology | CM-0530 | |
| Chemical compound, drug | Agarose | Aladdin Scientific | 11966311 | |
| Chemical compound, drug | Phosphate-buffered saline | Beijing DingGuo ChangSheng Biotechnology Co Ltd | BF-0011 | |
| Chemical compound, drug | Ultrapure water | ELGA Corporation | PURELAB flex system | |
| Software, algorithm | Origin | OriginLab Corporation | 2021 (RRID:SCR_014212) | |
| Software, algorithm | ImageJ | LOCI, University of Wisconsin | (RRID:RCR_003070) | |
| Other | Xenon lamp | Beijing zhongjiao Jinyuan Technology Co, Ltd | CEL-TCX250 | Light source |
| Other | Fiber optic spectrometer | Fuxiang Optics Co, Ltd | PG2000-Pro-EX | Instrument |
| Other | Microscope | Nikon | ECLIPSE Ti (RRID:SCR_021242) | Instrument |
| Other | D-LH/LC lamphouse | Nikon | | Light source |
| Other | 3D printer | Bambu Lab | P1P | Instrument |

### Design and fabrication of the side-view observation petri dish/device

The overall size of the device, as shown in *Figure 8A*, is approximately 17 mm in height, with a petri dish diameter of 35 mm. Inside the 3D-printed handle and the frame, two cylindrical magnets with a diameter of 3 mm and height of 3 mm were embedded (*Figure 8B*). To compare with the 35 mm petri dish, the height, width, and thickness of the first-surface mirror are 20, 4.5, and 2 mm (*Figure 8C*). A 3D printer (Bambu Lab P1P, China) was used to fabricate all the 3D-printed parts with a speed of 50 mm/s, the height of each slice is 200 µm, and infill density is 50%. Before coming into contact with cell samples, the device is first cleaned overall using 75% alcohol and then sterilized by UV light irradiation for 30 min. The total cost of manufacturing this device is less than $1.

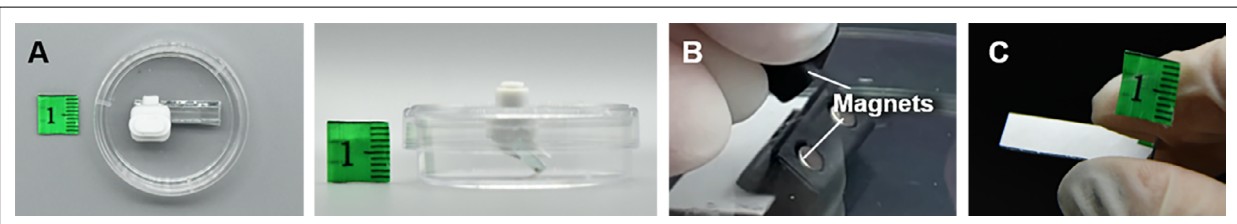

**Figure 8.** The display of the side-view observation petri dish/device. (**A**) The top- and side-view images of the side-view observation petri dish/device. (**B**) The magnets inside the device.

## Measuring the spectra of the DMEM culture medium and agarose

The optical properties of cell culture medium and agarose well for spheroids culture were characterized by fiber optic spectrometer. The optic fiber was mounted on an optical platform, with one end connected to PG2000-Pro-EiesX fiber optic spectrometer (Fuxiang Optics Co, Ltd, Shanghai, China) and the other to a computer for spectral measurements. The light source (CEL-TCX250 Xenon lamp) was placed at the same height as the fiber connection on the optical platform and preheated for 30 min. The 1 and 2 wt% agarose solution were prepared in advance, with 2 ml of each solution added to cuvettes and allowed to solidify for 15 min. During the absorption measurement, ddH$_2$O was first used to calibrate the control sample under the light source. Subsequently, the 1 and 2 wt% agarose block samples, as well as the DMEM medium, were measured sequentially. When directly measuring the spectra of the light source and the samples under the light source, the spectrum of the light source was measured first, followed by the 1 and 2 wt% agarose gel samples, and finally the DMEM medium.

## Experimental setting of the side-view observation of the 3D spheroids

To capture the side-view images of samples under an inverted microscope, we used the Nikon ECLIPSE *Ti* microscope (TS100, Nikon, Japan). The objectives we equipped are 4×, 10×, 20×, and 40× F objective lenses (Nikon, Japan) with *NA* of 0.13, 0.30, 0.45, and 0.60, respectively. The corresponding working distance of 4×, 10×, 20×, and 40× F objective lenses are 16.5, 15.2, 8.2–6.9, and 3.6–2.8 mm, respectively. The light sources we used to capture the bottom-view were D-LH/LC lamphouse (Nikon, Japan) and the light sources we used to capture the side-view was CEL-TCX250 (Xenon lamp light source, Beijing zhongjiao Jinyuan Technology Co, Ltd, China).

## Tracking the spheroids formation

The DU 145 cells were cultivated in agarose micro-wells using a previously reported procedure (*Pan et al., 2023a*). First, the agarose micro-well array was replicated from a 3D-printed mold. In brief, the replication process involved adding 700 µL of a 2% agarose solution to cover the 3D-printed mold, which was then allowed to solidify at room temperature (25°C). Once solidified, the agarose-based material was removed from the mold, resulting in an agarose micro-well array with a depth of 2 mm and a radius of 1 mm. The agarose micro-wells were then placed in the 35 mm petri dish. For tumor cell culture on the agarose micro-wells, the suspension with $2.5 \times 10^3$, $5 \times 10^3$, $1 \times 10^4$, and $2 \times 10^4$ DU 145 cells was added to each well. Subsequently, 600 µL of cell culture medium was added to cover the agarose base. The petri dish was placed in a cell incubator at 37°C with 5% CO$_2$. The bottom and side morphology of the spheroid formation was observed every 5 hr.

## Examining the spheroid migration from the side-view

$1 \times 10^4$ DU 145 cells/well in agarose micro-wells grown for 7 days were collected for migration assay. The spheroids were transferred into the petri dish with a side-view observation device installed on the lid. Then, the mirror was moved near the spheroid for observation using the handle on the lid. The petri dish was placed in a cell incubator at 37°C with 5% CO$_2$. The bottom and side morphology of the spheroid migration was observed every 12 hr.

## Observing the fusion of spheroids from different angles

$1 \times 10^4$ DU 145 cells/well in agarose micro-wells grown for 7 days were collected for fusion assay. In brief, spheroids were pipetted to the agarose micro-well which was placed in the cell culture petri dish. The mirror was attached to the agarose micro-wells by adjusting the handle at the lid of the petri dish. The petri dish was placed in a cell incubator at 37°C with 5% CO$_2$. The bottom and side morphology of the spheroid fusion process was observed over 48 hr.

## Evaluating the impact of the drug on the spheroid 3D structure

$1 \times 10^4$ DU 145 cells/well in agarose micro-wells grown for 3 and 7 days were collected for drug and NK-92 cells assay. In brief, spheroids were pipetted into the agarose micro-well which was placed in the cell culture petri dish.

### DOX

DOX was diluted with cell culture medium to 50 µg/mL and added to the micro-wells seeded with DU 145 spheroids. Then the bottom and side morphology of the spheroid was observed over 48 hr at 37°C with 5% $CO_2$.

### NK-92 cells

NK-92 cells were cultured in NK-92 special medium. For the co-culture assay, the NK-92 cells suspension was added to the micro-wells seeded with DU 145 spheroids. The petri dish was placed in a cell incubator at 37°C with 5% $CO_2$. The bottom and side morphology of the spheroid was observed over 48 hr.

## Statistics analysis

Data are expressed as mean ± SD of the number of biological replicates indicated in each figure legend. All the data analysis, bar graphs, and fitting curves were plotted by Origin (OriginLab Corporation, version 2023, USA).

## Acknowledgements

This work was financially supported by the National Natural Science Foundation of China (No. 32171401) and the Natural Science Foundation of Chongqing (CSTB2022NSCQ-MSX0808).

## Additional information

### Competing interests

Wen Sun, Yan Li: Affiliated to Sinopharm Animal Health Corporation Ltd. No other competing interest to declare. The other authors declare that no competing interests exist.

### Funding

| Funder | Grant reference number | Author |
|---|---|---|
| National Natural Science Foundation of China | 32171401 | Ling Yu |
| Natural Science Foundation of Chongqing Municipality | CSTB2022NSCQ-MSX0808 | Ling Yu |

The funders had no role in study design, data collection and interpretation, or the decision to submit the work for publication.

### Author contributions

Ke Ning, Conceptualization, Resources, Data curation, Writing – original draft, Writing – review and editing; Yuanyuan Xie, Data curation, Formal analysis, Methodology, Writing – original draft; Wen Sun, Resources, Visualization; Lingke Feng, Data curation; Can Fang, Conceptualization, Methodology; Rong Pan, Investigation, Methodology; Yan Li, Supervision, Project administration, Writing – review and editing; Ling Yu, Conceptualization, Supervision, Writing – original draft, Project administration, Writing – review and editing

### Author ORCIDs

Ke Ning http://orcid.org/0009-0007-6116-6134
Ling Yu https://orcid.org/0000-0002-6726-281X

Reviewer #1 (Public review): https://doi.org/10.7554/eLife.101886.3.sa1
Reviewer #2 (Public review): https://doi.org/10.7554/eLife.101886.3.sa2
Author response https://doi.org/10.7554/eLife.101886.3.sa3

## Additional files

**Supplementary files**
MDAR checklist

**Data availability**

The device has been uploaded to the public DRYAD database and can be used with your own 35 mm petri dish, magnets and first-surface mirror: https://doi.org/10.5061/dryad.9p8cz8wt7. When using, please refer to Figure 8A–C.

The following dataset was generated:

| Author(s) | Year | Dataset title | Dataset URL | Database and Identifier |
|---|---|---|---|---|
| Ning K, Xie Y, Sun W, Feng L, Fang C, Pan R, Li Y, Yu L | 2025 | Models from: Non-destructive in situ monitoring of structural changes of 3D tumor spheroids during the formation, migration, and fusion process | https://doi.org/10.5061/dryad.9p8cz8wt7 | Dryad Digital Repository, 10.5061/dryad.9p8cz8wt7 |

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
