## [Editor Report · eLife Assessment]

The ingenious design in this study achieved the observation of 3D cell spheroids from an additional lateral view and gained more comprehensive information than the traditional one angle of imaging. This extended the methods to investigate cell behaviors in the growth or migration of tumor organoids in a time-lapse manner and these extensions should be **important** to the field. The authors provide **compelling** evidence that the methods work as described.

---

## [Referee Report · Reviewer #1 (Public review)]

Summary:

The author developed a new device to overcome current limitations in the imaging process of 3D spheroidal structures. In particular, they created a system to follow in real-time tumour spheroid formation, fusion and cell migration without disrupting their integrity. The system has also been exploited to test the effects of a therapeutic agent (chemotherapy) and immune cells.

Comments on revised version:

The authors well addressed all my concerns. It is a wonderful design to view the 3D cell spheroids.

---

## [Referee Report · Reviewer #2 (Public review)]

Summary:

The author developed a new device to overcome current limitations in the imaging process of 3D spheroidal structures. In particular, they created a system to follow in real-time tumour spheroid formation, fusion and cell migration without disrupting their integrity. The system has also been exploited to test the effects of a therapeutic agent (chemotherapy) and immune cells.

Strengths:

The system allows the in situ observation of the 3D structures along the 3 axes (x,y and z) without disrupting the integrity of the spheroids; in a time-lapse manner it is possible to follow the formation of the 3D structure and the spheroids fusion from multiple angles, allowing a better understanding of the cell aggregation/growth and kinetic of the cells.

Interestingly the system allows the analysis of cell migration/ escape from the 3D structure analysing not only the morphological changes in the periphery of the spheroids but also from the inner region demonstrating that the proliferating cells in the periphery of the structure are more involved in the migration and dissemination process. The application of the system in the study of the effects of doxorubicin and NK cells would give new insights in the description of the response of tumor 3D structure to killing agents.

---

## [Author Response]

The following is the authors’ response to the original reviews.

**Reviewer #1:**
The ingenious design in this study achieved the observation of 3D cell spheroids from an additional lateral view and gained more comprehensive information than the traditional one angle of imaging, which extensively extended the methods to investigate cell behaviors in the growth or migration of tumor organoids in the present study. I believe that this study opens an avenue and provides an opportunity to characterize the spheroid formation dynamics from different angles, in particular side-view with high resolution, in other organoids study in the future.

Thank you for your positive response.

(1) Figure 1A and B, the images of "First surface mirror" are unclear. The authors should capture a single image of "First surface mirror" by high resolution. The corresponding information on the mirror should also be included in the manuscript.

Thank you for your kind reminder. To make the content more intuitive, we have added the clear image of the first surface mirror to Fig. 8C.

(2) The spheroids sizes in this study are 200-300 um. Whether this size is the limitation by the device? And which is the best size by the device? The size of spheroids suitable for this device should be characterized.

Thank you very much for your question. As shown in Fig. 1D, the imaging principle indicates that the sample size is theoretically not affected by the device. For larger biological samples or samples exceeding the size of a 35 mm petri dish, a larger container and first surface mirror can be used. However, in practice, it is not recommended to use this device with laboratory microscopes for samples exceeding 4 mm in size.

Firstly, the working distance of the microscope objective lens is limited by its factory specifications. Secondly, this device is designed to fit a 35 mm petri dish, and the first surface mirror can capture a maximum sample size of 4.5 mm. Fortunately, this size is more than sufficient for cell spheroids.

(3) Figure 2F. The scale bar covered the imaging and made it unclear. It was difficult to read and evaluate the quality of the images. And it seemed no obvious difference between 5 cm and 15 cm. Please carefully check this data.

Thank you very much for your question. First, we checked the image scale and coverage issues and made adjustments in the revised version. Secondly, when the light source was placed 5 cm from the sample, the sample itself appeared relatively clear, but the boundary with the background was less distinct. At a distance of 15 cm, the light source not only illuminated the sample effectively but also made the distinction between the spheroid and the background more apparent. To ensure consistency and stability in image capture, we ultimately selected a 15 cm distance between the sample and the light source for imaging.

(4) Figure 3A. It seemed that the seeding cells were initially located as a ring with a hole in the center. Why do not seed the cells evenly in the well?

Thank you very much for your question. First, the cells were added as a suspension, naturally settling at the bottom of the well during imaging. When seeded in agarose wells, the cells spontaneously aggregated over time, as shown in sVideo4. Our previous study showed that the use of agarose wells offers high fault tolerance and efficiency in cell spheroid culture (Pan, R. et al. Biofabrication, 2024, 16, 035016).

(5) I just wonder whether this design could be extended to the fluorescent imaging and how do it. Please give an expectation in the discussion.

Thank you very much for raising this key question regarding the imaging capability of this device. As shown in Author response image 1A, due to the specific nature of fluorescence imaging light sources, it is feasible to perform fluorescence imaging of cell spheroids using a microscope, including the built-in light source. Using 4′,6-diamidino-2-phenylindole (DAPI) staining, we captured fluorescence images of cell spheroids in both bottom-view and side-view modes (Author response image 1B), demonstrating that side-view observation of cell spheroids with this device is indeed feasible.

**Author response image 1. sa3fig1:** The device was used on an inverted microscope to capture fluorescence images. (A) Schematic of the imaging setup. (B) Bright-field and fluorescence images of cells. (Scale bar = 500 µm.)

(6) The first sentence in the introduction. "Three-dimensional (3D) spheroids" should be "Three-dimensional (3D) tumor spheroids".(7) P11, Line 7, "both lethal and lethal" should be corrected.(8) The writing and grammar should be polished.

Thank you very much for your suggestions to improve the quality of the article. We have made the necessary revisions in the updated version.

**Reviewer #2:**
Summary:The author developed a new device to overcome current limitations in the imaging process of 3D spheroidal structures. In particular, they created a system to follow in real-time tumour spheroid formation, fusion and cell migration without disrupting their integrity. The system has also been exploited to test the effects of a therapeutic agent (chemotherapy) and immune cells.Strengths:The system allows the in situ observation of the 3D structures along the 3 axes (x,y and z) without disrupting the integrity of the spheroids; in a time-lapse manner it is possible to follow the formation of the 3D structure and the spheroids fusion from multiple angles, allowing a better understanding of the cell aggregation/growth and kinetic of the cells.Interestingly the system allows the analysis of cell migration/ escape from the 3D structure analyzing not only the morphological changes in the periphery of the spheroids but also from the inner region demonstrating that the proliferating cells in the periphery of the structure are more involved in the migration and dissemination process. The application of the system in the study of the effects of doxorubicin and NK cells would give new insights in the description of the response of tumor 3D structure to killing agents.

We sincerely thank you for your detailed and supportive review of our manuscript. Your recognition of our system’s capabilities for in situ observation of 3D structures along multiple axes, as well as its potential applications in studying therapeutic effects, is highly encouraging. Your comments on the advantages of this system for analyzing cell migration, morphological changes, and responses to therapeutic agents are especially appreciated.

Thank you again for your thoughtful feedback and for highlighting the contributions of our work. Your insights have been invaluable in refining the focus and clarity of our study, and we hope that our revisions meet your expectations.